# PROPAGATOR: An Operational Cellular-Automata Based Wildfire Simulator

**Andrea Trucchia [1,\*]**, **Mirko D'Andrea [1]**, **Francesco Baghino [1]**, **Paolo Fiorucci [1]**, **Luca Ferraris [1,2]**, **Dario Negro [3]**, **Andrea Gollini [3]** and **Massimiliano Severino [4]**

[1] CIMA Research Foundation, I-17100 Savona, Italy; mirko.dandrea@cimafoundation.org (M.D.); francesco.baghino@cimafoundation.org (F.B.); paolo.fiorucci@cimafoundation.org (P.F.); luca.ferraris@cimafoundation.org (L.F.)
[2] Department of Environmental Engineering, University of Genova, I-16126 Genova, Italy
[3] Italian Department of Civil Protection, Presidency of the Council of Ministers, I-00189 Rome, Italy; Dario.Negro@protezionecivile.it (D.N.); Andrea.Gollini@protezionecivile.it (A.G.)
[4] Civil Protection Agency of Regione Lazio, I-00145 Rome, Italy; mseverino@regione.lazio.it
\* Correspondence: andrea.trucchia@cimafoundation.org

**Abstract:** PROPAGATOR is a stochastic cellular automaton model for forest fire spread simulation, conceived as a rapid method for fire risk assessment. The model uses high-resolution information such as topography and vegetation cover considering different types of vegetation. Input parameters are wind speed and direction and the ignition point. Dead fine fuel moisture content and firebreaks—fire fighting strategies can also be considered. The fire spread probability depends on vegetation type, slope, wind direction and speed, and fuel moisture content. The fire-propagation speed is determined through the adoption of a Rate of Spread model. PROPAGATOR simulates independent realizations of one stochastic fire propagation process, and at each time-step gives as output a map representing the probability of each cell of the domain to be affected by the fire. These probabilities are obtained computing the relative frequency of ignition of each cell. The model capabilities are assessed by reproducing a set of past Mediterranean fires occurred in different countries (Italy and Spain), using when available the real fire fighting patterns. PROPAGATOR simulated such scenarios with affordable computational resources and with short CPU-times. The outputs show a good agreement with the real burned areas, demonstrating that the PROPAGATOR can be useful for supporting decisions in Civil Protection and fire management activities.

**Keywords:** wildfire; cellular automata; stochastic simulators

## 1. Introduction

### 1.1. Wildfire as a Menacing Natural Hazard

Mediterranean countries are particularly prone to wildfires, which represent a significant menace to environment, properties, and human lives. Even in countries aware of the fire danger conditions and well equipped for firefighting, there is still a lack of prevention and preparedness capacities in order to deploy in a short time all the activities able to share among the first responders and Civil Protection Authorities (CPAs) the main information to cope with direct impacts on exposed people. The tragic wildfires that occurred in Greece and in Portugal in the last few years, which caused many fatalities [1], and the more recent event that occurred last summer in Gran Canaria, where 9000 people were evacuated [2], constitute examples of the consequences of such shortcomings.

Most of the wild-land fires in the Mediterranean are human caused; however, natural ignitions caused by lightning are not negligible and could be increased by climate change [3]. Human-caused

fires result from many different reasons, ranging from campfires left unattended to stubble burning, negligently discarded cigarettes, or intentional acts of arson. In addition, wild-land fires can be ignited by anthropic elements such as power lines and railways. Wildfire emergencies, especially in the southern EU Countries, are related with extreme weather conditions, characterized by persistent dry strong winds over flammable land cover species [4]. In this case, the ignition probability increases and, in case it happens, the fire propagation is rapid and difficult to cope with: in most of the recent wildfire emergencies, casualties happened in few hours after the fire ignition. For this reason, it is extremely urgent to support first responders and CPAs with operational tools in emergency response, based on reliable wildfire risk maps and efficient emergency plans. This behavior requires CPAs to improve their ability of anticipation, discrimination, and selection of the best strategies and the most appropriate decisions in the first phase of the event, in order to ensure security to exposed people.

Prevention, preparedness, and fire fighting activities usually involve several operational structures and different stakeholders, which need to be well informed and coordinated. New technologies and computer modeling represent a great opportunity, supporting wildfire emergency managers sharing information useful in the coordination of civil protection and fire fighting activities. Specifically, evacuation and traffic management represent the main issue to save lives. The recent dramatic events occurred in Greece and Portugal (such as the mega-fires of June and October 2017) made evident the need of tools able to anticipate the behavior of fire in order to implement prevention and communication activities in time to save lives. This can be achieved using ad hoc mathematical and numerical models, like the one described in the present work, or implementing some other kind of decision-making process, as the spatial segmentation into polygons of fire potential introduced by Castellnou et al. in [5]. Such schematics account for operational opportunities of different complexity levels, and may help to keep the main focus on the information needed to make decisions by reducing noise from maps and simulations.

*1.2. Mathematical Modeling: An Ally in Wildfire Management*

Unfortunately, physical processes influencing wildfire propagation are complex, meaning that the effects of slopes, wind conditions and fuel moisture interconnect and combine together, determining the evolution of the fire event. Such factors make wildfires multi-scale, multi-physics, and nonlinear phenomena. This makes the formulation of efficient and reliable mathematical models particularly hard, as well as their computational implementation.

Nevertheless, in literature, there are many different approaches and models dedicated to this specific task. Such modeling efforts are usually divided into three main approaches [6–10]:

1. empirical and semi-empirical models, which rely on statistically derived laws of fire propagation [11,12];
2. macroscopic-deterministic models, where the fire spread is modeled in a continuum, mainly by using computational fluid dynamics techniques coupled with atmospheric, heat transfer, and combustion models [13–16];
3. stochastic lattice or grid-based models, where the evolving quantities are usually described adopting a discretization in space and time, and dealing with the propagation of the fire front from a cell to the neighboring ones by adopting detailed localized evolution rules that comprehend the underlying physics at the desired level of resolution [7,17–21].

In any case, it should be remembered that the distinction between such categories may not be strict as expected, since, in many works, different approaches are mixed together [7,8,22].

Moreover, it is common knowledge that any of the aforementioned modeling framework may or may not be the right one for the specific task intended by the practitioner. Every modeling approach is in fact characterized by strengths and drawbacks. To begin with, empirical models are quite straightforward to implement and use, and do not require a high computational budget. They have proved to approximate, under certain restrictions due to their simplified nature, fire-spread dynamics

in an acceptable way. However, most such models are derived from controlled laboratory experiments, and their reliability in predicting fires that took place under different conditions and landscapes is to be questioned [9].

The second category, macroscopic-deterministic models, are mostly based on first principles. However, since they try to model intrinsically nonlinear, multi-scale, stochastic, and complex phenomena, their prediction accuracy cannot be always ensured. Moreover, these models typically need high computational resources for simulations on large-scale heterogeneous areas, and-or the computational time is not comparable to the simulated time, making such model not suitable for tactical intervention scenarios.

Grid-based stochastic modeling techniques may thus fill the gap between the first two formulations. Such techniques approximate the complex and inherent stochastic underlying physics, grasping the very mechanisms of fire spread by describing via probabilistic methods the physics at the microscopic/local scale. The front propagation at the macroscopic scale emerges as the result of the rules operating at the detailed (local) level. They are often lightweight models, versatile in the sense that they can be integrated in the framework of existing databases with relative ease. At the cost of some preliminary modeling, these models can: (i) at the physical level incorporate both theoretical/first principles and (semi-) empirical relations inside of the probabilistic mathematical core; and (ii) easily integrate spatial and temporal heterogeneity of the initial and boundary conditions of the simulation, i.e., dealing with spatial patterns of the vegetation type, orography, meteorological conditions. They can easily be coupled with Geographical Information Systems (GIS) and take (possibly real-time) meteorological data as input [7].

Adopting grid-based stochastic modeling techniques, also more complex fire propagation patterns, such as the fire spotting effect, can be also simulated in a rather straightforward way via ad hoc probabilistic rules that may make use of fire intensity, wind field, and fuel characteristics [23].

Cellular Automatas (CA) constitute one of the most well known examples of the latter category of models ([24,25]).

CA models for wildfire simulation model discretize spatial interactions by adopting a square or hexagonal [26] grid. The macroscopic fire spread dynamics is simulated by the means of an ensemble of different realization of a stochastic process. In every realization, the spreading of the fire front from burning cells to neighboring ones is modeled by the means of probabilistic rules. Although CA models may simplify the underlying physical processes, their modular nature allows them to reach the desired level of complexity and accuracy. This can be achieved under more accurate physical modeling and-or employing state-of-art numerical algorithms. For what concerns the first path, in [27], a CA model has been coupled with existing forest physical models to ensure better accuracy of fire spread simulation. On the other hand, Ghisu et al., in [18], provided elaborate CA models that overcome typical constraints imposed by the shape of the grid and may perform comparably to deterministic models such as FARSITE [28], requiring, however, higher computational budget.

*1.3. The Synopsis of* PROPAGATOR *Implementation: History of the Development of an Operational and Easy-To-Use Simulator*

The implementation of PROPAGATOR has spanned across more than a decade, and it is recapitulated in the following paragraphs. Robustness, ease of implementation, and quick operational deployment have always been the beacon during the overall process. The conceptual road-map is portrayed in Figure 1.

The first implementation of PROPAGATOR started from a request of the Italian Civil Protection Department, to support the organization of the G8 summit 2009, originally planned in La Maddalena, Sardinia, a region frequently affected by severe forest fires in summer season in order to evaluate the best prevention measures and support the fire fighting activities in case of a forest fire event. It had been equipped with a Google Web Toolkit based interface, with a server written in MATLAB$^{\circledR}$ whose code was running all concurrent processes as a unique large process (letter *a*) of Figure 1).

Even though the implementation would have been improved in the following years, for the first time, it was possible to run fire propagation simulations all over Italy, from a simple web interface. Since its first release, it was able to reproduce burned areas up to 10,000 ha in a few minutes of computational time. Given an ignition point, it highlighted the zones more likely to be affected by fire propagation, while no temporal iso-countour nor propagation speeds were given as output. Since no fuel moisture content was implemented in the model, the output in this version has been used as a *worst-case scenario*, highlighting the most endangered areas given conditions of totally dry fuel (letter *b*) of Figure 1).

In 2011, the second release of PROPAGATOR was operational. In this release, the model has been implemented in a 3D environment named NAZCA (letter *c*) of Figure 1). While in the previous version algorithm and server code were mixed, at this stage, the algorithm was running as a standalone MATLAB® script, much easier to maintain and to develop. The model was shipped as a plugin, together with the data-set of fuel cover and DEM of the whole Italy. This was still quite cumbersome and the object of further improvements. Timing algorithms were added at this stage, taking into account wind and orography. Probability maps were at last time dependent and isochrones are added as visual output of the model (letter *d*) of Figure 1). At this stage, PROPAGATOR did not include a real parametrization of the propagation speed (and thus gave no information on the absolute time scale of the overall process). In 2014, the third release was completed (letter *e*) of Figure 1). The web interface was redesigned and it was integrated into the multi purpose MyDewetra platform [29], a tool for the forecasting, monitoring and real-time surveillance of all the environmental risks (http://www.mydewetra.org).

The simulations could be run also on Lebanese territory owing to the results of the international project "*Establishment of Sustainable Natural Resources Management Platform and Early warning system*" implemented for the National Council for Scientific Research (CNRS-Lebanon), in the framework of the "*Regional Coordination on Improved Water Resources Management and Capacity Building*" initiative, funded by World Bank/GEF.

In 2017, the fourth release saw a total rewriting of the code in the Python programming language, with a new server stack, with Django REST API and database, and Celery task dispatcher (letter *f*) of Figure 1). Some of the algorithms for the treatment of slope and wind data have been rewritten from scratch, and it has been made possible to change wind conditions over time. An open API had been released to several developers during the ANYWHERE (*EnhANcing emergencY management and response to extreme WeatHER and climate Events*) European Project [30] (letter *g*) of Figure 1) and the fuel—DEM dataset had been extended from the sole Italian territory to Finland, Portugal, Spain (Catalonia, Cantabria, Asturias) France (Corsica and Cote D'Azur) and Switzerland. Figure 2 portrays the regions in the Europe and Mediterranean basin where PROPAGATOR simulations can be run using Dewetra platform and-or the ANYWHERE open API.

In 2020, the fifth release of PROPAGATOR (letter *h*) of Figure 1) saw the implementation of a Rate of Spread (RoS) model in order to give the isochrones a more realistic time parametrization, and the introduction of the fuel moisture into the computational core. The 2020 version also saw the introduction of fire fighting actions (lines and polygons where some kind of fire fighting procedure is going to be put in charge) that may be prescribed by the user in a time-dependent way. The Python3 code has been substantially rebuilt according to Object Oriented Programming procedures, in order to rise its modularity for further improvements. In the standalone version, there is the opportunity to furnish directly the data for DEM and fuel cover, in order to launch simulation of any part of the globe, provided that there is available field data. Running within the Mydewetra platform, meteorological Numerical Weather Prediction (NWP) models have been integrated in PROPAGATOR providing synoptic data for wind direction and magnitude. Dead Fine Fuel Moisture Content model data [31] have been integrated as well into the platform.

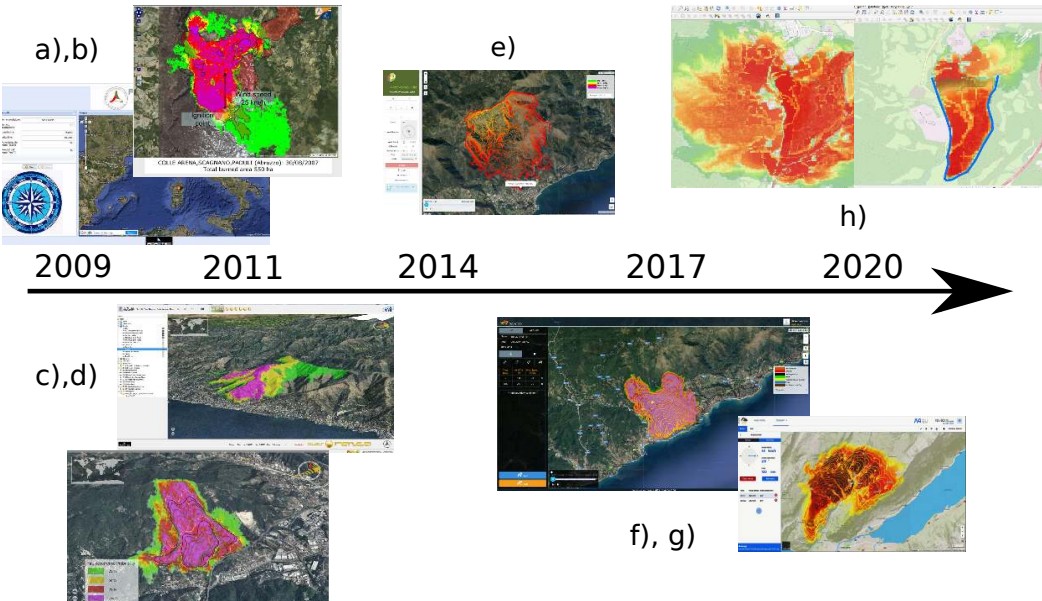

**Figure 1.** The roadmap of operational `PROPAGATOR` implementation, from 2009 to 2020. Each letter represents a milestone in the development of the project, explained in detail in the main text.

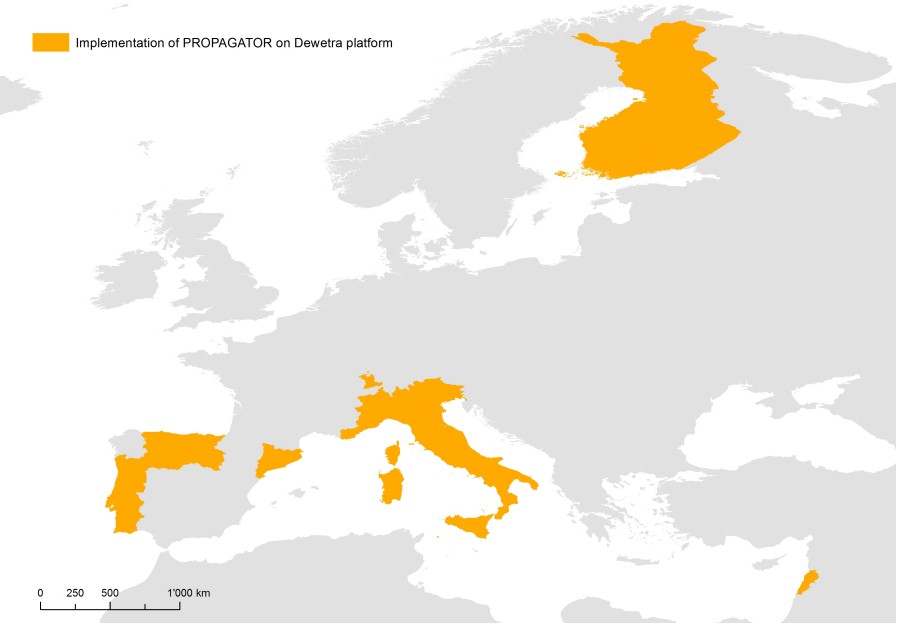

**Figure 2.** All the regions were `PROPAGATOR` simulations can be run using a Dewetra platform.

## 2. `PROPAGATOR` **Model**

The `PROPAGATOR` model is a quasi-empirical stochastic CA model based on a raster implementation, which discretizes the space into a grid composed of square cells of arbitrary length $\Delta x = \Delta y = L$. The cell size reflects the resolution in space of the analysis and the final results. In this work, $L$ has been fixed to 20 m, allowing `PROPAGATOR` to give high resolution output, fundamental for reproducing the middle-sized Mediterranean fires object of the following sections. At the center of the cells, information on the elevation and vegetation cover are interpolated from input rasters (Digital Elevation Model and land-cover raster files).

For each time step, every cell is characterized by a state taking values from a finite set. More specifically, each cell of the domain can assume one out of three different possible states:

- State 1 corresponds to cells that are *burning* during the current simulation step;
- State 0 corresponds to cells that are *already burned* in previous steps of the simulation;
- State $-1$ corresponds to cells that are *unburned*, but that can burn in the following steps of the simulation.

The possible changes in the state of the cells are: an unburned cell can become a burning cell with a probability that depends on the parameters of the simulation, or it can remain unburned; a burning cell becomes a burned cell after one time step with probability equal to 1; a burned cell can not change its state.

When the fire propagates from a cell to another, the latter is given a specific time for the fire front to completely cross it, described more specifically in the following parts. When this time elapses, the cells switches its state from unburned $(-1)$ to burning $(1)$, and it will then try to propagate fire to the adjacent unburned cells. In operational terms, the computed time step $\Delta t$ for the state change is appended accordingly to a scheduler which manages the fire propagation mechanism of the cellular automaton.

The state diagram of this automata is depicted in Figure 3.

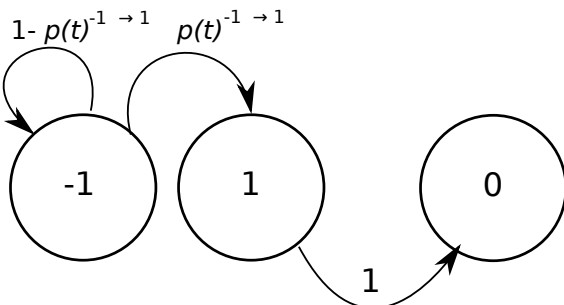

**Figure 3.** The state diagram of the automata adopted in `PROPAGATOR`. States $-1, 1, 0$ stand for unburned cell, burning cell and burned cells, respectively. At a given time $t$, an unburned cell has a probability $p^{-1\rightarrow1}$ to burn. Such probability is given by the overall state of the stochastic realization, initial, and boundary conditions. At the subsequent time step of the stochastic process, every burning cell is going to be set to a burned cell (and thus inactive).

The fire propagation is modeled as a contamination process between adjacent cells of the considered domain; the probability of fire spreading from a cell to one of its neighborhood, $p_{ij}$, is calculated starting from the nominal fire spread probability (named $p_n$ in the following), which is then modified considering several factors. Such factors account for the topography, wind vector, and the fuel moisture content. In addition, the evolution in time of the fire is modeled by combining the nominal fire spread velocity ($v_n$ in the following) and the same influencing factors, by the means of a convenient implementation of an RoS model. For each cell of the simulation, corresponding to a point $x_P = (x, y)$ of the spatial domain, the model calculates the probability $u(x_P, t)$ of being burnt at time $t$ and space $x$ evaluating the fire frequency for each cell, based on a significant number of stochastic simulations and each simulation is performed for the same ignitions and wind conditions. Throughout this work, a number $N = 100$ of realization has been adopted. This procedure is resumed in Figure 4.

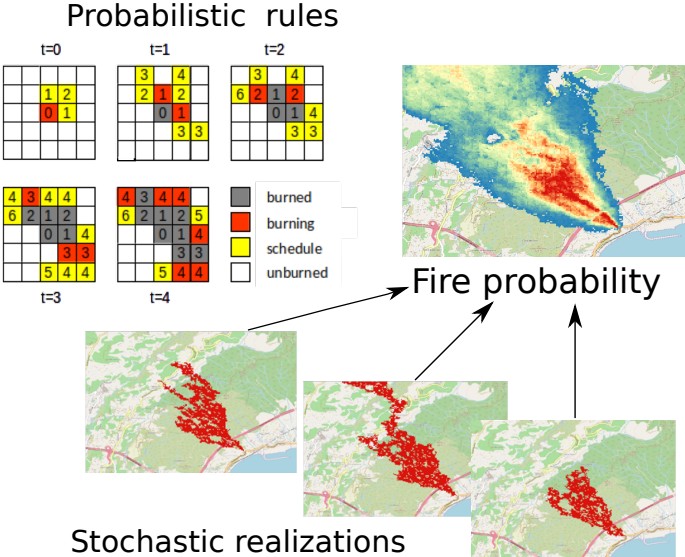

**Figure 4.** Averaging procedure of single realization adopted in `PROPAGATOR`.

The cellular automata is applied on the Moore neighborhood, a two-dimensional square lattice composed of a central cell, the *i*-cell that is the ignited one, and the eight cells that surround it, as shown in Figure 5, which can be ignited by the *i*-cell. The fire spreading is stochastically calculated considering the directions between the center of the *i*-cell and the ones of the neighboring cells, the slopes between the cells and the possible different moisture conditions. Each cell is characterized by a vegetation type.

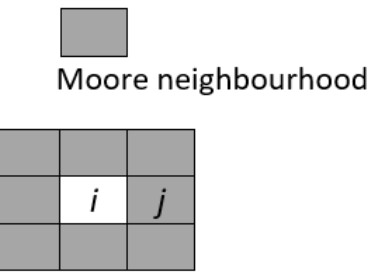

**Figure 5.** Moore neighborhood implemented in `PROPAGATOR`.

Fuel models are adopted by widespread fire propagation models ([28,32]) to classify the physical characteristics such as fuel load, heat content, and height of live and dead biomass that contribute to the size, intensity, and duration of a fire [33]. In literature, several practical guides and collections of standard ([33,34]) and custom-local fuel models (see, e.g., [35–37]) are available. However, in order to preserve ease of use and portability, `PROPAGATOR` adopts a manageable simplified custom fuel model with seven available fuel types corresponding to seven different types of vegetation. The considered fuel types are the following: broad-leaves, shrubs, grasslands, fire-prone conifers, agro-forestry areas, non-fire prone forest, and non-vegetated areas.

The class called "non-vegetated areas" includes man-made buildings and infrastructures (e.g., streets, villages and towns) and the non-vegetated terrains, such as natural bare soil. Fire propagation cannot take place in this class. Rivers, lakes, and seas are considered by default as non-burnable areas as well.

The fire propagates from a cell *i* to the neighbor cell *j* with a probability $p_{ij}$, called Fire Spread Probability, which depends heavily on the involved vegetation types. The $p_{ij}$ is also influenced by the slope between the two cells, the wind effect (direction and velocity), and the fuel moisture content of

the *j*-cell. The probability of the fire propagation $p_{ij}$ from an ignited *i*-cell at the time $t_k$ to a *j*-cell is calculated applying the cumulative binomial probability formula [38], Equation (1):

$$p_{ij} = (1 - (1 - p_n)^{\alpha_{wh}}) \cdot e_m \tag{1}$$

where $p_n$ is the nominal Fire Spread Probability, $\alpha_{wh}$ is the factor that combines the topographic and wind influence on the probability and $e_m$ is the factor that simulates the effect of the fine fuel moisture content.

The model takes into account the vegetation of the cell that is burning and the cells where the fire can propagate and it analyzes how a certain type of vegetation can ignite other types of vegetation, or also the same vegetation type. These probability values are given in input through a fire spread probability table, Table 1, which considers all the possible combinations between the different vegetation and land-cover types. The nominal Fire Spread Probability $p_n$ represents the possibility for the *i*-cell, characterized by a certain vegetation cover, to ignite an adjacent *j*-cell, characterized by the same, or another, vegetation cover.

**Table 1.** In the first six rows, the values of the nominal fire spread probability $p_n$ between all the species are given. In the last row, nominal fire spread velocity $v_n$ is reported.

| | | Burning Cell | | | | | |
|---|---|---|---|---|---|---|---|
| | | Broadleaves | Shrubs | Grassland | Fire-Prone Conifers | Agro-Forestry Areas | Not Fire-Prone Forest |
| neighbor cells | Broadleaves | 0.3 | 0.375 | 0.25 | 0.275 | 0.25 | 0.25 |
| | Shrubs | 0.375 | 0.375 | 0.35 | 0.4 | 0.3 | 0.375 |
| | Grassland | 0.45 | 0.475 | 0.475 | 0.475 | 0.375 | 0.475 |
| | Fire-prone conifers | 0.225 | 0.325 | 0.25 | 0.35 | 0.2 | 0.35 |
| | Agro-forestry areas | 0.25 | 0.25 | 0.3 | 0.475 | 0.35 | 0.25 |
| | Not fire-prone forest | 0.075 | 0.1 | 0.075 | 0.275 | 0.075 | 0.075 |
| **Nominal Fire Spread Velocity [m/min]** | | 100 | 140 | 120 | 200 | 120 | 60 |

"Not fire-prone forest" class represents the low-flammable forests: its probability of being ignited is quite low, except if the burning cell is a "Fire-prone conifers" cell. Medium flammable tall vegetation is considered in the "Broadleaves" class, while "Fire-prone conifers" class included the highly flammable tall vegetation. There are also three classes which represent the medium to low vegetation: "Agro-forestry areas" represent areas with a low vegetable density characterized by low probability of propagation; the "Shrubs" class includes the medium-flammable low vegetation; "Grassland" class represent the high-flammable very low vegetation.

The $p_n$ values were defined according to a continuous and thorough calibration through all the development of the model, and valuable information deriving from fire susceptibility mapping [39].

The slope and the wind speed and direction can modify the initial value of $p_n$, increasing or decreasing the nominal value depending on the direction of propagation. The influence of the topography is taken into account through the slope between the two cells. The slope increases the propagation probability $p_n$ when the slope increases in the direction of propagation (uphill case) and it decreases $p_n$ if slope decreases in the direction of propagation (downhill case).

The topographic influencing factor $\alpha_h$ is defined as shown in Figure 6.

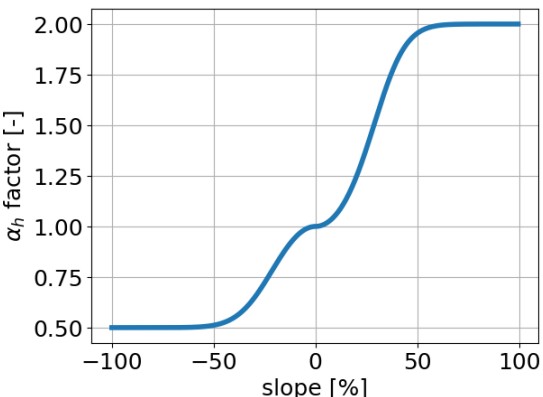

**Figure 6.** Topographic influencing factor $\alpha_h$.

The wind influence is taken into account by considering the wind velocity and its direction (both considered to be homogeneous in the domain) relative to the direction from the burning cell and the adjacent ones. At every time step, the value of wind for every cell is perturbed in both magnitude and direction. Wind magnitude is modified by applying a multiplicative uniform noise $\mathcal{U}[0.8, 1.2]$, where $\mathcal{U}[a, b]$ stands for the uniform distribution with support in $[a, b]$. The wind direction is perturbed adopting a uniform additive noise $\mathcal{U}[-11.25°, 11.25°]$. The wind influencing factor $\alpha_w$ is shown in Figure 7. The influence is significant only if wind speed is quite high: in the low-speed case, it does not modify the probability of propagation, not increasing nor decreasing. However, when the wind speed is sufficiently high, it has a big impact on the probability of propagation. The wind direction plays a key role in the overall process because the probability of burning is increased when the propagation direction is aligned with the wind direction, while it is decreased when the directions are opposite.

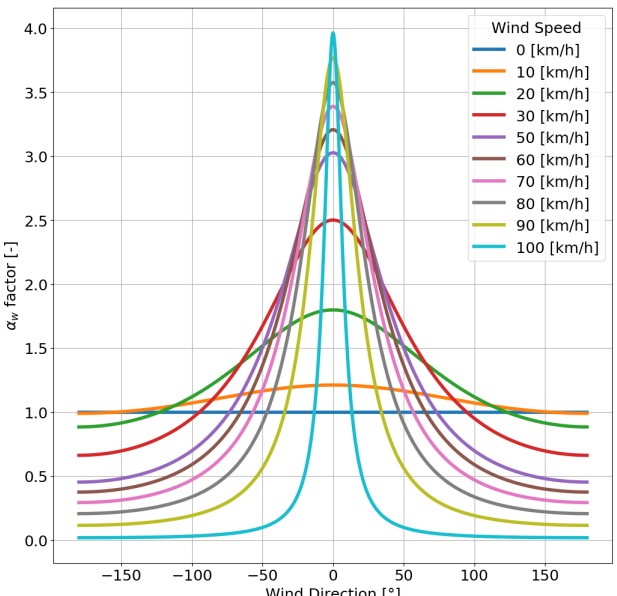

**Figure 7.** Wind influencing factor $\alpha_w$.

The factor that combines the topographic influence with the wind influence on the Fire Spread Probability, named $\alpha_{wh}$, is obtained as follows:

$$\alpha_{wh} = \alpha_w \cdot \alpha_h. \tag{2}$$

In Figure 8, it is possible to notice how $\alpha_{wh}$ impacts the Fire Spread Probability, $p$: when $\alpha_{wh}$ is equal to 1, the nominal transitional probability is obtained; when this factor is not unitary, it is possible to evaluate the effect of the possible combinations of slope, wind speed, and direction. The Fire Spread Probability can thus vary in a range between at about zero and 0.7, a value that in numerical experiments makes propagation quite always possible, see again Figure 8.

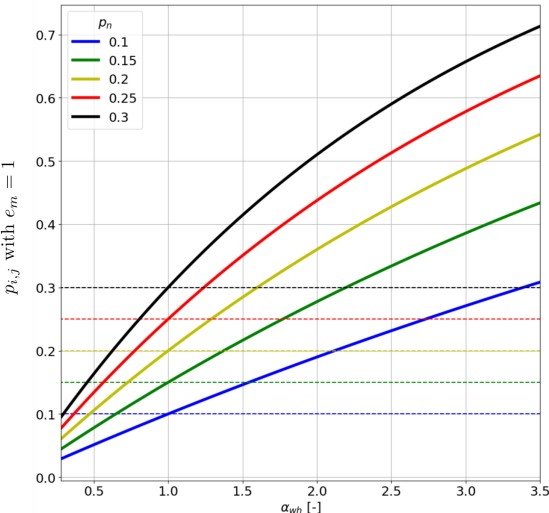

**Figure 8.** Influence of the combined slope-wind factor on the Fire Spread Probability. The plot portrays the dependence of $p_{ij}$ of Formula (1) on the slope-wind factor $\alpha_{wh}$, given a fixed $e_m = 1$ and several values for $p_n$.

The effect of fuel moisture content on Fire Spread Probability has been modeled as a factor $e_m$, which multiplies the nominal probability $p_n$. This factor has been implemented as described by Burgan and Rothermel in [32], and is portrayed in Figure 9. It ranges in the interval $[0, 1]$ and it is computed as a function of the fuel moisture ratio, i.e., the dead fuel moisture over the dead moisture of extinction. For the sake of simplicity, the value of 0.3 has been used for the moisture of extinction for any of the vegetation classes considered in `PROPAGATOR`, applying a conservative estimate [34].

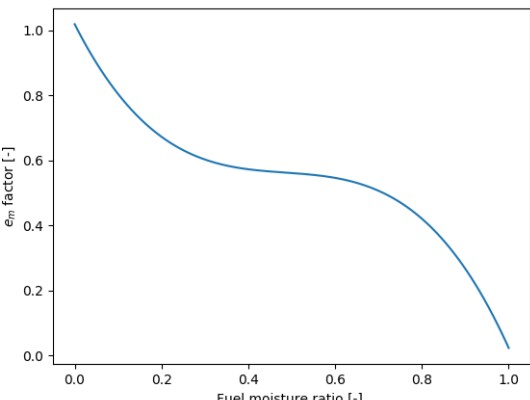

**Figure 9.** Effect of the fuel moisture ratio (dead fuel moisture over dead moisture of extinction) on the Fire Spread Probability.

When a cell is ignited, the transition time of the fire is modeled by combining the Rate of Spread $v_{prop}$ and the fuel moisture factor $f_m$, with the distance $d$ from the center of the $i$-cell which propagated the fire to the center of the newly ignited $j$-cell. In particular, the transition time $\Delta t$ is calculated as:

$$\Delta t = \frac{d}{v_{prop} \cdot f_m}.$$ (3)

It is acknowledged that the flammability of the vegetal fuel and, consequently, the rate of the spread of a fire depends exponentially on the fuel moisture content, see, e.g., [7]. $f_m$ is calculated using the formulation proposed by Marino et al. [40]:

$$f_m = e^{c \cdot M_n}$$ (4)

where $c$ is a constant that has been set at $-0.014$ and $M_n$ is the fuel moisture (ranging from 0 to 1).

The Rate of Spread $v_{prop}$ is calculated starting from the nominal $v_n$, which stands for the Fire Spread Velocity for each vegetation type without slope and wind effects, and then modifying it by considering the slope and the wind effects. Values for $v_n$ are reported in Table 1. Slope and wind effects have been evaluated through the formulations proposed by Sun et al. [21], which are calculated as Equations (5) and (6).

The wind speed factor $K_w$ is evaluated as:

$$K_w = exp(0.1783\,V)$$ (5)

where $V$ is the wind velocity in the direction of propagation, in [m/s].

The slope factor $K_\phi$ is evaluated as:

$$K_\phi = exp(3.533\,(tan\phi)^{1.2})$$ (6)

where $\phi$ is the terrain slope angle $[°]$ in the direction of propagation.

The Rate of Spread $v_{prop}$ is then evaluated multiplying the nominal Fire Spread Velocity $v_n$ by the two factors, $K_w$ and $K_\phi$, as proposed by Sun et al. [21].

Since PROPAGATOR involves several input factors of different physical nature, different preliminary tests are required to see if the response to the set of input factors is reasonable. An initial response analysis to variation in vegetation type, vegetation spatial pattern, slope, wind speed, wind direction, and fuel moisture content is given in the Supplementary Materials. Please note that, in the context of homogeneous fuel distribution, the response analysis to fuel moisture content may also be considered a response analysis to variation of the base $p_n$ value.

Fire fighting actions have been implemented distinguishing operations that use water lines from other, heavier operations that act on the vegetation, such as interventions made using earth-moving machines. The first actions are implemented setting the fine fuel moisture content at a prescribed value $\mu_{wl} = 0.8$ in the domain cells where fighting actions are enforced, so that the fire propagation probability is strongly reduced; heavy equipment actions are implemented changing the values of the vegetation type at "Non-vegetated areas" where these actions are modeled, so that fire can not propagate in those areas. In the proposed test cases of Section 3, the first type of Fire Fighting action is tested.

## 3. Case Studies

For that work, PROPAGATOR has been tested on five wildfire events that occurred in Italy (two events) and Catalonia, Spain (three events). They have been selected because they represent different types of wildfires: the Catalan events are characterized by large (and well described) impact of the fire fighting activities; the Italian ones were developed in areas where human actions have a

very low impact, fire was mainly driven by vegetation, orography and environmental conditions (and the fighting actions were not recorded).

*3.1. Data Retrieval*

3.1.1. Ignition Point, Wind Speed, and Wind Direction

The general data of the simulated fire events are shown in Table 2. In the table, "Latitude" and "Longitude" are expressed in WGS84 CRS; the "Burnt area" (expressed in hectares) is the area burnt during the actual events; "Wind speed and direction" are taken as significant for the entire fire event, while, in the simulation, more detailed time dependent wind conditions have been implemented; the level of "Human activity" means how fire fighting actions influenced the final shape of the fire event: "Low" level means that the fire event was mainly driven by vegetation, orographic, and environmental conditions and the simulated human activities are very low or null; "Medium" level means that the fighting actions focused on little areas of the wildfire and influenced limited part of the simulation; "High" level means that the impact of fighting actions is significant in large parts of the fire events and the final shape is strongly influenced by these actions.

**Table 2.** Data of the case studies simulated in this work with `PROPAGATOR`.

| Fire Event | Region (Nation) | Date | Ignition Point | | Burnt Area [ha] | Wind Speed [km/h] (Direction [°]) | Human Activity |
|---|---|---|---|---|---|---|---|
| | | | Latitude | Longitude | | | |
| Avinyo | Catalonia (Spain) | 05/07/2017 | 41.83733 N | 1.97016 E | 90 | 10 (180) | High |
| Blanes | Catalonia (Spain) | 24/07/2016 | 41.70457 N | 2.77539 E | 30.6 | 35 (200) | High |
| Fasce mountain | Liguria (Italy) | 06/09/2009 | 44.39118 N | 9.03743 E | 945.3 | 45 (50) | Low |
| Ittiri | Sardinia (Italy) | 23/07/2009 | 40.57170 N | 8.58768 E | 5130.7 | 40 (240) | Low |
| Sant Fruitos de Bages | Catalonia (Spain) | 22/07/2017 | 41.73432 N | 1.86608 E | 105.2 | 20 (120) | Medium |

For what concerns Catalan wildfires, the data regarding ignition points, wind speed, and directions were retrieved from the web archives of the Ministry of Home Affairs (*Departament D'Interior*) of the Government of Catalonia http://interior.gencat.cat/ca/arees_dactuacio/bombers/foc-forestal/incendis_forestals/informes-dincendis-forestals/. On the other hand, the data related to wind speed and direction of the Italian events were recorded by available anemometers located closer than 10 km from the burnt area (see panels (a) and (b) of Figure 11), belonging to the sensor network of the Italian Civil Protection Department (http://www.protezionecivile.gov.it/). In the case of Ittiri wildfire, because of the distance between the ignition point and the used anemometer, several simulations have been carried out considering different wind directions and wind magnitudes, also making use of NWP models computed in the closest grid point to the wildfire ignition. The results of this sensitivity study are reported in the Supplementary Materials. The ignition point for the Ittiri wildfire has been retrieved from the official dataset of Regione Autonoma della Sardegna, while the ignition point for theMonte Fasce wildfire has been retrieved from the official report 'Foglio AIBFN' provided by the Italian State Forestry Corps, *Corpo Forestale dello Stato* (now *Carabinieri Forestali*).

### 3.1.2. Fire Fighting Actions

The fighting actions that have been simulated follow for Catalan fires the descriptions recorded on the reports produced by the Catalan Fire-Fighters in the aforementioned web archives of the Goverment of Catalonia. For what concerns the Italian cases, fire fighting actions are located along secondary streets where major human intervention is supposed to have been deployed. In the specific case of the Ittiri fire, fire fighting actions are taken from [41].

### 3.1.3. Burned Area Geometries

The shape-files concerning burnt areas for the Catalan fires are retrieved from the web archives of the Department of Agriculture, Hunting, Fishing and Food of the Catalan Government (http://agricultura.gencat.cat/ca/serveis/cartografia-sig/bases-cartografiques/boscos/incendis-forestals/incendis-forestals-format-shp/). The actual Italian wildfires areas are recorded on the online platform MyDewetra.

### 3.1.4. Land-Cover Files

In Catalan fires, the Land Cover Map of Catalonia (MCSC) has been adopted (http://www.creaf.uab.es/mcsc/usa/descriptiu.htm). Such map is a high resolution thematic cartography of the main types of land cover of the country (forests, crops, urban areas, etc.). The MCSC is carried out in the Centre de Recerca Ecològica i Aplicacions Forestals (CREAF), with the funding of the Generalitat de Catalunya. It is a 241 categories map with a minimum resolution of 10 m.

Land-cover in Liguria was retrieved from the 240 categories 1:25,000 maps available at the geographical web portal of Ligurian Region https://geoportal.regione.liguria.it/catalogo/mappe.html

Land-cover in Sardinia was provided by the Autonomous Region of Sardinia's archives, by the means of a 1:25,000 map with 70 land use categories.

All of the aforementioned maps have been rasterized into a 7-category, 20-meter resolution map, as explained in Section 2.

### 3.1.5. Orography Files

For the simulation of Catalan wildfires, a 20-m resolution DEM input file provided by the Catalan Ministry of Home Affairs has been adopted, while, for the Italian wildfires, the 20-meter resolution map made available by the Italian Institute for Environmental Protection and Research ISPRA has been used (http://www.sinanet.isprambiente.it/it/sia-ispra/download-mais/dem20/view).

## 4. Results

In order to show the predicting capabilities of the presented model, PROPAGATOR has been tested on the wildfires described in Section 3. The simulator run on a 4 core (2.5 GHz) AMD A10-9620P laptop with 8 GB of RAM. The CPU time was lower than the simulated physical time by several orders of magnitude, as shown in Table 3. The first column corresponds to the time duration of the results shown in Figures 10 and 11, while the second one reports the time duration in minutes of each simulation.

Figure 10 portrays the comparison between the actual burnt area and the simulated fire probability distribution for the Catalan wildfires of Avinyo (panel (a)), Blanes (panel (b)), and Sant Fruitos de Bages (panel (c)), while, in Figure 11, the results of Fasce (panel (c)) and Ittiri (panel (d)) wildfires are reported. From a qualitative point of view, the iso-contours of the fire probability distribution is in line with the final shape of the wildfires, for both Catalan and Italian cases. Nevertheless, the whole spatial distribution of the output probability (the color shading of Figures 10 and 11) can shed some light on different wildfire scenarios. This is particularly true for the Catalan wild-land fires, where the spatial distribution of the fire arrival probability also embraces the cases when the fire fighting is not effective in containing the fire spread (panels (b) and (c) of Figure 10).

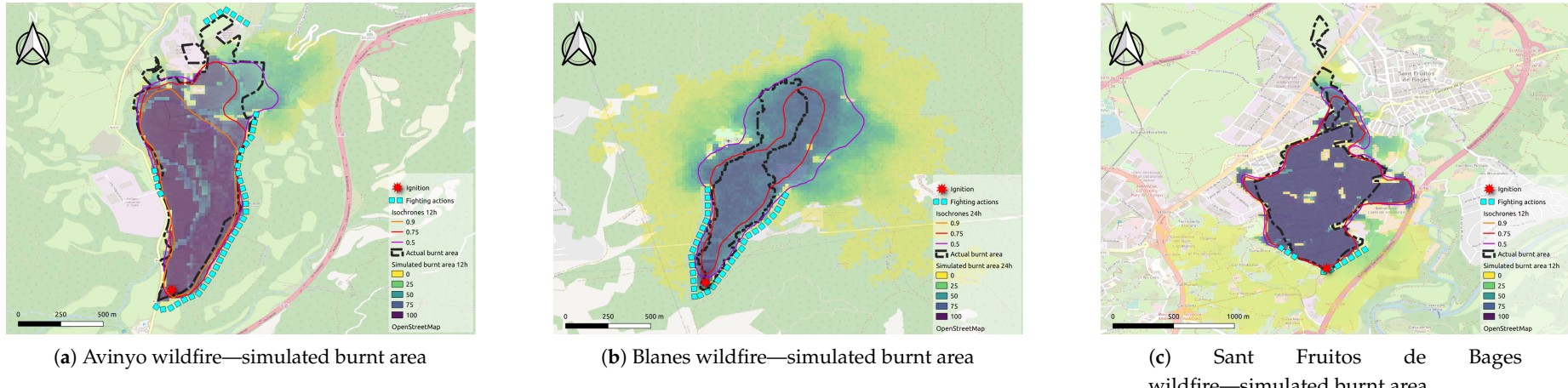

(**a**) Avinyo wildfire—simulated burnt area

(**b**) Blanes wildfire—simulated burnt area

(**c**) Sant Fruitos de Bages wildfire—simulated burnt area

**Figure 10.** Comparison between the actual burnt areas and the simulated ones for the Catalan wildfires. The three images (**a**–**c**) show the actual burnt areas (black dash-dot lines), the ignitions (red stars), the implemented fighting actions (light blue dashed lines), the simulated burnt areas with their color scale, and the isochrones (colored lines) produced at the end of the simulations, which represent the burnt areas with probability higher than 50%, 75%, and 90%.

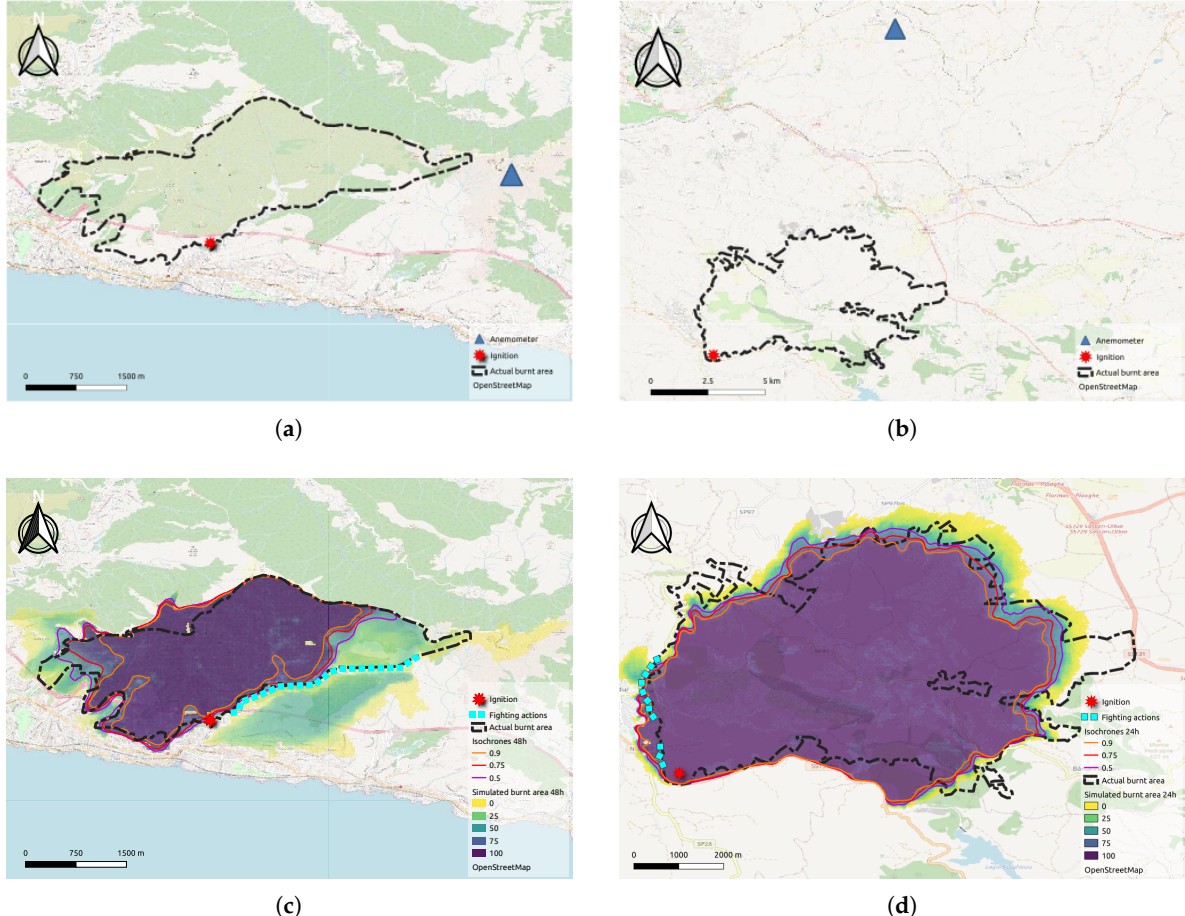

**Figure 11.** First row: the location of the anemometers used for Fasce Mountain wildfire (**a**) and Ittiri wildfire (**b**) is represented by triangular glyphs. Second row: the two images show actual burnt areas (black dash-dot lines), the ignitions (red stars), the implemented fighting actions (light blue dashed lines), and the simulated burnt areas with their color scale and the isochrones (colored lines) produced at the end of the simulations, which represent the burnt areas with probability higher than 50%, 75%, and 90% for Fasce Moutain (**c**) and Ittiri (**d**) fires.

In order to show the effect of the fire suppression actions on the extent and topology of the simulator output, simulations with and without the fire fighting actions have been performed. In Figure 12, the results are shown with respect to the Avinyo case study. It is evident from the picture that an accurate representation of fire fighting actions has a remarkable impact on the simulation output, both on the displayed probability iso-contours and the overall spatial distribution of PROPAGATOR 's output.

While the results are shown for the final time step of the simulations, PROPAGATOR output is time-dependent and the temporal evolution of the probability iso-contours can be also displayed. Figure 13 shows such time-dependent evolution for the Ittiri test case, considering the 50% probability isocontours.

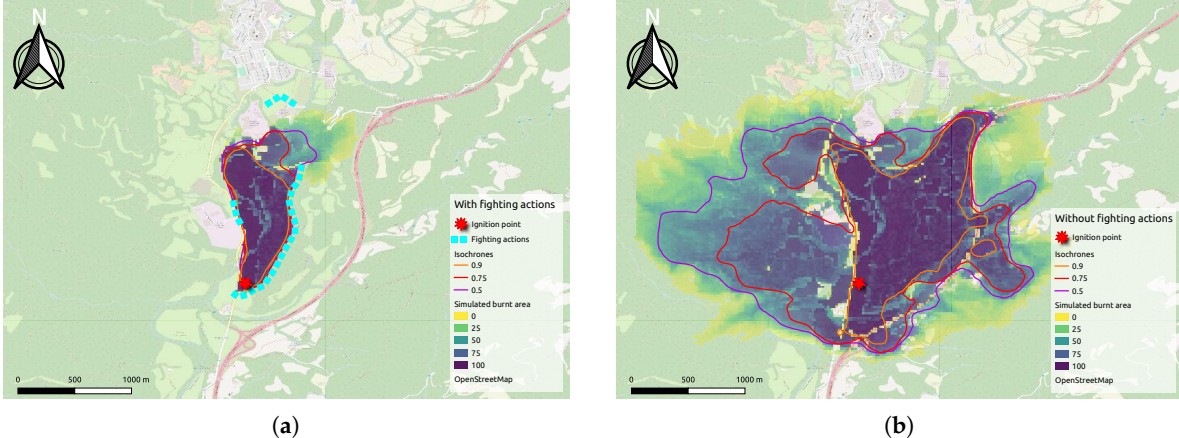

(**a**)　　　　　　　　　　　(**b**)

**Figure 12.** Comparison between the results obtained implementing the fighting action (**a**) and without this implementation (**b**) for the Avinyo test case. The ignition point is represented with a red star, while the fighting actions are in light blue dashed line; the simulated burnt areas are shown both representing the probability of being burnt (colored scale) and the isochrones that represent the burnt area with a probability higher than 50%, 75%, and 90% after 12 h of simulation.

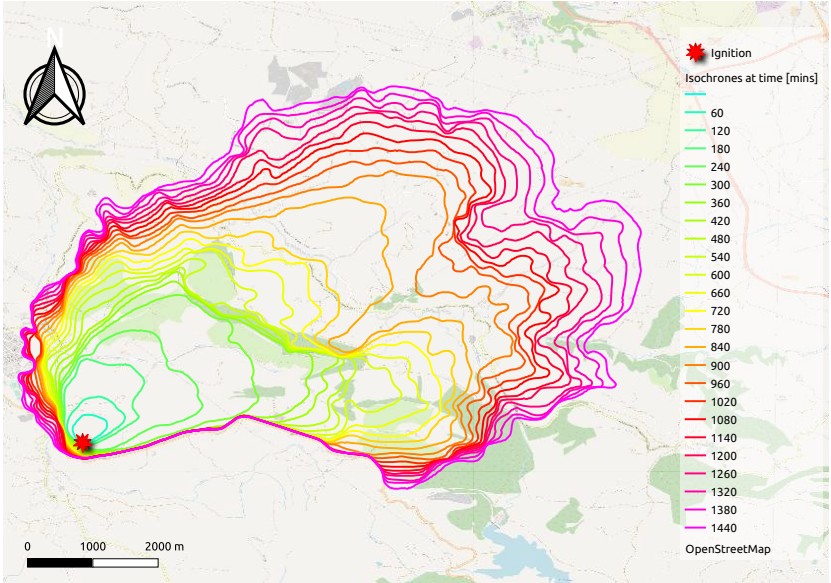

**Figure 13.** Time history of the simulation of Ittiri wildfire. The colored lines represent the isochrones that identify the simulated burnt areas with probability higher than 50% that are produced at each time step (one hour) until the end of the simulation (24 h).

### 4.1. Performance Indicators

In order to have an estimate of the performance of a `PROPAGATOR` run, three ad hoc performance indicators ($I_i$, $i \in \{1, 2, 3\}$) have been considered, as well as four standard performance indicators, namely the Sorensen coefficient, the McNemar test, the sensitivity, and specificity.

This notation for the confusion matrix components [42] corresponding to the predictive capabilities of `PROPAGATOR` when compared to the observed burned area will be of use in this section. First, when evaluating a performance indicator, a computational cell is considered to be burned by `PROPAGATOR` when the computed probability of fire arrival $u$ is greater than a threshold, set to 0.5. The first entry for the confusion matrix, $a$, is the number of cells coded as burned in both observed and simulated fires; $b$ is the number of cells that did not burn in the actual fire but were considered burned by `PROPAGATOR`; $c$

is the number of cells that burned in the actual fire but were considered as unburned by `PROPAGATOR`. The aforementioned performance indicators are briefly described below:

### 4.1.1. Indicator $I_1$

The Euclidean norm computed over the the whole computational domain $\mathcal{S}$ between the indicator function of the real fire burnt area $\mathcal{I}_b(x_P, T_{\text{end}})$, that returns 1 inside of the real burnt area and 0 outside, and the probability output of `PROPAGATOR`, $u(x_P, T_{\text{end}})$, scaled by the measured burnt area $|\Omega_{\text{burnt}}|$:

$$I_1 = \frac{1}{|\Omega_{\text{burnt}}|} \sqrt{\int_{\mathcal{S}} |\mathcal{I}_b(x_P, T_{\text{end}}) - u(x_P, T_{\text{end}})|^2 dx_P}, \tag{7}$$

### 4.1.2. Indicator $I_2$

The Euclidean norm computed over the whole computational domain $\mathcal{S}$ between the indicator function of the real fire burnt area $\mathcal{I}_b(x_P, T_{\text{end}})$ and the area with simulated burning probability $u$ at time $T_{\text{end}}$ higher than 0.5, described by the indicator function $\mathcal{I}_{>0.5}(u(x_P, T_{\text{end}}))$, scaled by measured burnt area $|\Omega_{\text{burnt}}|$:

$$I_2 = \frac{1}{|\Omega_{\text{burnt}}|} \sqrt{\int_{\mathcal{S}} |\mathcal{I}_b(x_P, T_{\text{end}}) - \mathcal{I}_{>0.5}(u(x_P, T_{\text{end}}))|^2 dx_P} \tag{8}$$

Since the probability of fire arrival $u$ ranges in the compact interval $[0, 1]$, there is not a unique way along which the simulation and the real burnt area may coincide or not for a given computational cell. Indicator $I_1$ accounts for this richness of information of the probabilistic output, integrating for every point the squared distance between the probability of arrival and the binary information of the real burnt area raster (burned-unburned, represented by the indicator function $\mathcal{I}_b$). On the other hand, the indicator $I_2$ considers the probabilistic output $u$ as a binary indicator after applying a threshold.

### 4.1.3. Indicator $I_3$

The area of the measured final burnt area where `PROPAGATOR` did not forecast correctly the fire arrival $u(x_P, T_{\text{end}}) < 0.5$, scaled by the measured burnt area $|\Omega_{\text{burnt}}|$. $I_3$ can be thought as a *false negative* estimator, and its minimization is of paramount importance in the case that a *worst case scenario* is investigated:

$$I_3 = \frac{|\mathcal{S}^*|}{|\Omega_{\text{burnt}}|}, \tag{9}$$

where $\mathcal{S}^*$ is the part of the burnt domain given by the relation

$$x \in \mathcal{S}^* \iff \mathcal{I}(x_P, T_{\text{end}}) = 1 \text{ and } u(x_P, T_{\text{end}}) < 0.5.$$

The goal value for the three $I_i$ indicators, $i \in \{1, 2, 3\}$ is zero.

### 4.1.4. Sorensen Coefficient $S_c$

The Sorensen similarity coefficient, or Sorensen similarity index ($S_c$) [43] is a statistical index which computes the value of similarity between two samples. $S_c$ values were calculated as follows:

$$S_c = \frac{2a}{2a + b + c} \tag{10}$$

The result is a value between 0 and 1, where 1 means a perfect agreement between observation and simulation, and 0 means that there is no agreement.

Since $S_c$ does not involves areas that are unburned for both observations and simulations, its value is not affected by the extent of the simulation domain $\mathcal{S}$.

### 4.1.5. McNemar Test

The McNemar test [41,42,44] is a non-parametric test based on the chi-square ($\chi^2$) distribution that evaluates the symmetry of rows and columns of the confusion matrix related to actual and simulated burned areas.

In the following, the corrected version of the McNemar test is adopted [45]:

$$\chi^2 = \frac{(|c - b| - 1)^2}{(c + b)} \tag{11}$$

The McNemar test was performed to evaluate the null hypothesis of marginal homogeneity between actual and simulated burned areas.

### 4.1.6. Sensitivity and Specificity

The sensitivity, or producer's accuracy, [41,46] is the ratio between cells correctly classified as burned by `PROPAGATOR` and the total number of actually burned cells; its complementary measure (1-sensitivity) stands for the probability of committing a false negative (error of omission). Conversely, the specificity, or user's accuracy, [41,46] is the ratio between the number of cells correctly classified as unburned by `PROPAGATOR` and the total number of actually unburned cells. Its complementary measure, that is (1-specificity), corresponds to the probability of committing a false positive (error of commission).

The values of the performance indicators are tabulated in Table 4. $I_2$ is always greater than $I_1$. This could suggest that using all the probabilistic output $u$ of `PROPAGATOR` instead of focusing only on a selected isocontour somehow enriches the simulation capabilities of the proposed method, catching better the overall fire dynamics. Since `PROPAGATOR` has been initially conceived as a *worst case scenario* forecaster, the small values assumed by $I_3$ for any considered fire (including the one neglecting fire fighting actions) is not surprising. Given that $I_i$ indicators, $i \in \{1, 2, 3\}$ are scaled through the final burnt area, their values for the proposed Italian fires, which are much larger than the Catalan ones, are generally lower. However, this does not necessarily apply for Sorensen index, Sensitivity, and Specificity, where this distinction in performance indicators for the two groups of fires is not observed. The values of the McNemar $\chi^2$ test show that all the considered case studies exhibited a significant association ($P = 0.01$) between actually burned areas and simulated burned areas.

As expected, the test case of Avinyo without fire fighting (second line of Table 4) performed significantly worse than its counterpart with the correct representation of suppression activities (first line of Table 4). However, since the simulation without fire fighting actions covered a much wider area, (see Figure 12), it exhibits better indices for what concerns $I_3$ and Sensitivity. This motivates the inclusion of a complete set of performance indicators, capable of taking into account both omission and commission errors.

**Table 3.** Time duration of the simulations.

| Wildfire | Time Duration of the Simulation [h] | Time Duration of CPU Time [min] |
|---|---|---|
| Avinyo | 12 | $\simeq 1$ |
| Blanes | 24 | $\simeq 2$ |
| Fasce mountain | 48 | $\simeq 5$ |
| Ittiri | 24 | $\simeq 5$ |
| Sant Fruitos de Bages | 12 | $\simeq 1$ |

**Table 4.** Performance of the simulation of the proposed test cases.

| Wildfire | $I_1$ [−] | $I_2$ [−] | $I_3$ [−] | $S_c$ | McNemar $p$-Value | *Sens.* | *Spec.* |
|---|---|---|---|---|---|---|---|
| Avinyo | $1.37 \times 10^{-2}$ | $1.51 \times 10^{-2}$ | $1.48 \times 10^{-1}$ | 0.81 | $<10^{-2}$ | 0.85 | 0.96 |
| Avinyo (no Fire Fighting) | $5.05 \times 10^{-2}$ | $6.05 \times 10^{-2}$ | $1.19 \times 10^{-1}$ | 0.22 | $<10^{-2}$ | 0.88 | 0.63 |
| Blanes | $2.95 \times 10^{-2}$ | $3.63 \times 10^{-2}$ | $2.05 \times 10^{-1}$ | 0.63 | $<10^{-2}$ | 0.98 | 0.93 |
| Fasce | $3.08 \times 10^{-3}$ | $3.39 \times 10^{-3}$ | $1.66 \times 10^{-1}$ | 0.86 | $<10^{-2}$ | 0.83 | 0.96 |
| Ittiri | $1.24 \times 10^{-3}$ | $1.34 \times 10^{-3}$ | $1.06 \times 10^{-1}$ | 0.89 | $<10^{-2}$ | 0.89 | 0.92 |
| Sant Fruitos de Bages | $1.19 \times 10^{-2}$ | $1.34 \cdot 10^{-2}$ | $1.10 \times 10^{-1}$ | 0.82 | $<10^{-2}$ | 0.89 | 0.96 |

## 5. Discussion

The results obtained in the previous sections generated interesting insights of different nature that are worth discussing.

To begin with, the results highlighted that the current state of PROPAGATOR may be capable of helping the end users in the difficult task of emergency response. This can be stated because of the very low computational budget needed, the fast results given by the CA, and the good overall performance with the tested fires. Understanding uncertainty is one of the best approaches to improve capacities in natural disaster emergency management. A fundamental issue is whether it is better to get uncertain but reliable information on the fire front in a few seconds using very low computational resources or instead pursuing a very detailed output with elapsed CPU time comparable with the time scale of the considered event, at the cost of a sensibly higher computational effort. The proposed work derives directly from such question. When a fire is ignited, which area will probably be most affected? Is the fire front spreading towards an urban area? Is the fire front going to affect a road menacing people? Only having such information in time, however uncertain, it would be possible to implement prevention activities, such as people displacement, traffic network management, and any other activity designed to prevent harm to the population. With an increase in prevention and preparedness capacities, the disasters that every year afflict many regions of the world may be avoided.

One of the strong points of PROPAGATOR that emerged in the proposed research is that it relies on relatively easy-to-obtain data in order to launch meaningful simulations. The stochastic nature of its mathematical core allows for a simpler handling of boundary conditions: in particular, since the wind data do not need to be re-scaled, no additional computational load is generated. Regarding the integration of moisture data, it is worthwhile to spend a few words to further discuss it. Initially, PROPAGATOR was developed to provide only the worst case scenario, considering dead fine fuel completely dry. One of the most demanding feature from end users, in the last few years, has been the inclusion of the real-time conditions of fuel moisture corresponding to the ignition time, in order to simulate the different behavior of a potential fire depending on the real time and antecedent weather conditions. Besides PROPAGATOR, the authors developed and continuously update and maintain RISICO, a fire danger prediction model, for the Italian Civil Protection Department and other regional and international end users [31,47,48]. Such model is based on the evaluation of the hourly dynamic of the Dead Fine Fuel Moisture Content, considering both the fuel type and the effect of weather, in terms of Total precipitation, Relative Humidity and Air Temperature (2 m), and Wind Speed (10 m). Thus, every day, RISICO provides several runs using in input weather observations and forecasts. Thanks to the MyDewetra platform, the PROPAGATOR application can access the near real-time dead fine fuel moisture content in the area where the ignition is defined and its evolution during the simulation time. Shifting the attention from the input set to the output, the probabilistic output of PROPAGATOR allows for a richer information content in the results given to the end users. This is the case when the output probability field $u$ does not exhibit a sharp drop from 1 to 0, i.e., when the results show larger areas characterized by intermediate levels of fire arrival probability. This additional information may

be of use, for example, when the operational users have to visualize the effect of firebreaks for fast decision-making processes.

Regarding the significance of the proposed set of test cases, it needs to be remarked that the tested fires are not significant by their number (since, for the sake of simplicity, only five examples have been reported) but because of their characteristics: three smaller Catalan fires are examined, where plenty of data were available to model fire fighting actions; two Italian fires are examined as well, one characterized by a downhill propagation of the fire and the other one, wind driven, significantly larger than the others. On top of that, it goes without saying that reaching this level of qualitative and quantitative prediction dealing with a heterogeneous set of wildfire events work that, reaching this point, has involved a continuous honing of the algorithm, which took place at every release of `PROPAGATOR`. Such continuous improvement process implied dealing with hundreds of wildfire occurrences, which varied significantly in a vegetation pattern, burnt area, wind conditions, orography, and human intervention level.

Another key point that needs to be discussed is the empirical nature of several parameters such as the Nominal Fire Spread Probability and Nominal Fire Spread Velocity, and the functions related to wind and slope effects that modified the probability of fire propagation. Such functions have been an important part of the continuous improvement process mentioned before. One of the addressed questions is how good simple principles may perform, which are partly taken from existing literature, and partly derived empirically via confrontation with experts and end users. However, the proposed research aims at showing a starting point rather than presenting the perfect recipe for simulating wildfire occurrences with a CA approach. As a final observation, the current overall context of the paper is clearly not a basic research on fire propagation processes, rather than an attempt to pave the way to science technology transfer into operation. The proposed approach has been developed trying to include the main aspects that need to be considered and that may be estimated in real time during emergency, providing a comprehensive tool useful for end users. The aim of this paper is not testing the CA method in detail with extensive benchmarks and a thorough sensitivity analysis, but instead to spread the proposed approach among the community and provide the described tool to potential interested users. `PROPAGATOR` has been already tested operationally by several end users during the ANYWHERE [30] demonstration phase, and it will be operationally used from the next summer season by the Italian National Fire Corps in Liguria, within the Interreg Marittime—MEDCOOPFIRE [49] project.

## 6. Conclusions

In this paper, an operational wildfire simulator designed to respond to a broad scenario of wildfires, via an ad-hoc implementation of CAs and Rate of Spread, is described. The model is applied to a set of Spanish and Italian case studies, which range from fires that were controlled by a high level of human intervention to wild-land fires that lacked such level of fire fighting. When available, opportune information about fire fighting procedures has been considered in the simulations.

The proposed test cases thus represented ideal scenarios to test a CA model designed for wind-driven wildfires. Such CA has been equipped with a physical core that accounts for the effects of wind speed and direction, slope, vegetation pattern, fuel moisture content, and fire fighting actions. The simulation of the wildfires has been made using a probabilistic approach based on ensembles of $N = 100$ simulations that allowed estimation of the probability of burning of a given cell by evaluating the mean value of the fire arrivals on the given cell. The probabilistic output of `PROPAGATOR` gave useful information for the fire managers, since fast and reliable information about locations of burning probability is of paramount importance for allocating and deploying resources as well as for fire fighting.

The results have been validated via several fitness indicators, and the good overall performance obtained confirms the success of more than a decade of continuous scientific and technical improvement of an operational tool conceived for practitioners and decision-makers.

The principal lessons learnt that originated from the proposed work are reported in the following. First, the previous releases of the presented simulator gave acceptable results in very heterogeneous topographic and vegetation conditions where the burned area is mainly determined by such conditions. However, in practical scenarios, wind, fine fuel moisture and fire fighting actions play a key role on rate of spread and fire behavior: this stimulated further research. End users need to know in advance not only the potential area impacted by the events, but also information on the temporal evolution of the fire front. This led to the inclusion in `PROPAGATOR` of fire front propagation speed, and to the implementation in the simulator of fire fighting actions. Another feature that originated from operational users requests was the introduction of fuel moisture effect, to discriminate potential extreme events from easily controllable ones. In order to consider and test the latest improvements, the presented case studies regarded two wind driven cases (the Italian wildfires) and three other cases (the Catalan ones) that have been characterized by a higher degree of human intervention. In the latter cases, the introduction of fire fighting actions in the model improved drastically the overall accuracy of the model, given the condition that accurate information on the deployed actions is available. For what concerns the correct reproduction of wind driven fires, the interaction between complex topography and wind conditions in empirical CAs constitutes a topic of complex frontier research that will surely need some effort in this sense. In the wind driven test cases, the final burned area predicted by `PROPAGATOR` is highly sensitive to the time-dependent wind input data, as also suggested by the sensitivity analysis of wind conditions for the Ittiri test case available in the Supplementary Materials. Shifting the attention from the time resolution of the input wind data to the spatial one, it is remarked that the information of wind has not been downscaled, thus avoiding the use of computationally intensive and time consuming techniques. This did not prevent `PROPAGATOR` from giving acceptable results for the presented test cases. On the overall scope of the proposed approach, it should be emphasized that, when the final purpose is not the one of performing a real-time simulation of a chaotic physical system in extreme conditions to guide tactical actions of Fire Fighters, a fast CA approach that relies on empirical approach can deliver useful probabilistic scenarios of the fire front evolution. Such results may be helpful for Civil Protection decision makers and fire managers. Owing to the probabilistic nature of the output, the results of this stage of `PROPAGATOR` are suitable for decision-making processes delivering fast scenarios with time dependent probability of fire front arrival.

However, the modeling work done with `PROPAGATOR` is far from complete. Among the improvements planned for the future releases of `PROPAGATOR` , the first one in the implementation of long range fire spotting effects. Other important steps will be a Global Sensitivity Analysis procedure on the input parameters [50], and a re-scaling of the model, changing spatial and temporal resolution in order to simulate fires larger than 10,000 ha. The study of the impact of different fuel models on the simulation output is also planned.

It should be mentioned that, even though the produced research on `PROPAGATOR` can be considered an interesting preliminary study with a promising capacity of reproducing the fire behavior in a variety of situations, it still lacks a precise quantitative validation framework. To overcome this issue, an automated framework of systematic model testing and validation is going to be presented in future works, possibly coupled to parameter calibration procedures and the aforementioned Global Sensitivity Analysis.

**Supplementary Materials:** The response analysis of the model is available online at http://www.mdpi.com/2571-6255/3/3/26/s1: Figures S1–S7, Table S1.

**Author Contributions:** Conceptualization, A.T., P.F., M.D., D.N., and A.G.; Writing—original draft, A.T. and F.B.; Writing—review and editing, A.T., F.B., P.F., D.N., and A.G.; Investigation, A.T., M.D., and F.B.; Supervision, P.F., D.N., A.G., and M.S.; Software, M.D.; Funding Acquisition, L.F. All authors have read and agreed to the published version of the manuscript.

**Funding:** This research was funded by Italian Civil Protection Department—Presidency of the Council of Ministers through the convention with CIMA Research Foundation. This research was partially funded by the Horizon 2020 ANYWHERE project (grant agreement No 700099).

**Acknowledgments:** The authors acknowledge the Italian Civil Protection Department—Presidency of the Council of Ministers, who funded this research through the convention with the CIMA Research Foundation, for the development of knowledge, methodologies, technologies, and training, useful for the implementation of national wildfire systems of monitoring, prevention, and surveillance. The research activity presented in this paper was partially funded by the Horizon 2020 ANYWHERE project (grant agreement No 700099). Special thanks are due to Bernardo De Bernardinis who had the idea of a model to fulfill the need of a simple, yet operational tool to support civil protection activities. The Authors are also grateful to Gianfilippo Micillo for stimulating discussions and suggestions due to his great experience in forest fire management. Special thanks are also due to the ANYWHERE Team, in particular to its program manager Daniel Sempere Torres for his stimulating encouragement in advancing research and new operational implementations. Moreover, the Authors would like to thank all the stakeholders for their useful suggestions, Emanuele Gissi and all the WUIFI Team, who stimulated further research for the future development and improvement of the model. Finally, the Authors are grateful to the four anonymous reviewers for making the manuscript improve in readability and completeness.

**Conflicts of Interest:** The authors declare no conflict of interest.

## Abbreviations

The following abbreviations are used in this manuscript:

CA　　　Cellular Automata
RoS　　Rate of Spread
DEM　　Digital Elevation Model
NWP　　Numerical Weather Prediction

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
