# Peer review of "PROPAGATOR: An Operational Cellular-Automata Based Wildfire Simulator"

_fire, doi:10.3390/fire3030026_

Round 1

Reviewer 1 Report

The paper is very well written and provides description of the new version of the PROPAGATOR. The materials and methods are well described. The only question that I have is computation of the fuel moisture content. Does this parameter depend on weather conditions? What climatic variables are used to calculate this parameter?

In my view, this information is missing in the current manuscript, and this would be great if the authors could fix that.

Author Response

The Authors

Reviewer 2 Report

This paper details an implementation, deployement and testing of a simulation code based based on the CA. Propagation is simulated by the means of probabilities in each cells, and parameterized in a mostly empirical way (which is the same way Rothermels model is built, so pragmatic approach)l. The paper is well written I and I have no issues with the content and presentation of the paper or figures.

The current overall context of the paper is essentially software engineering (the connection of the web interface to the pre-existing solver, the testing of system, files in input). Although technically of a good quality, It is clearly not a basic research paper (a paper testing the CA method in details with more benchmark and sensitivity analysis would be that) but as the tool is public, this publication is important for potential interested users.

I recommend publication as it is.

Author Response

The Authors

Reviewer 3 Report

The subject of the article is related to the selected model of the spread of external fire and its application to forest fires that have taken place in recent years in Italy and Catalonia (Spain). It is becoming more and more important due to the need to predict threats from the ever-growing number of forest fires caused primarily by global warming and, consequently, by longer periods of drought.

Although the topic is interesting, the authors did not go too deep into the topic, as evidenced by the practical lack of discussion and conclusions in chapter 5. There is only a summary that is mostly a repetition of previous chapters and future prospects. It is necessary to supplement this chapter with an analysis of the results obtained and formulate the most important general conclusions (it is best to number them). For this purpose, you can use fragments of the texts contained in Chapter 4. Conclusions should be placed in a separate chapter

General remarks

Line numbers are written in too small font that does not match the template. In many places of the article no passive voice was used, which should be the rule in technical works. Abbreviations should be placed at the beginning of the article or explained in the text when they were first used

In addition, the layout of the work content is incorrect in many places. This applies especially to the organization of drawings and tables and their location in the text.

Detailed comments in chronological order are given below:

Line 157 - instead of “iso-chrone”, it should be “isochrone”.

Line 116 - Figure 2 (numbering to 1) should be placed before Figure 1 immediately after line 116, in which its description is placed. This drawing can be slightly reduced, because the text on it is invisible anyway and put it as standard across the page (it will take up less space). The description of individual letters appearing in the drawing should be transferred from the caption under the drawing to the main text.

Line 154 – instead of “Figure 1: should be “Figure 2” (figures should be renumbered).

Line 164 – Figure 1 should be placed on page 4 (the small drawing currently covers more than one page).

Line 174 – instead of “3” should be “three”.

Line 181 - It is not explained in the paper why after one time the burning cell will be burned, whether this step is constant or variable. If constant, then the burning out times of individual cells are different depending on many factors. It would be good to explain it here.

Line 189 - The location vector should not be marked with the same letter as its x coordinate.

Line 205 – instead of “7” should be “seven”.

Lines 217-219 - Explanations of physical quantities should be placed directly under formula (1)

Lines 220 – 226 + 234-235 - The text contained between these lines should be placed before Table 1.

Table 1 and Figure 6 – No sources are given after the titles, where the values listed in Table 1 as well as figures 6 and 7. If they are original, this has not been clearly stated. If not, please provide source. There was insufficient justification as to why these and no other values were used

Line 251 - The designation of the size appearing on the left side of the formula (2) differs by a subscript 0 from the value appearing in the formula (1). Are they the same or different sizes. If different, this has not been sufficiently explained. If the same, they should have the same designation.

Line 263 – The description of the size fm under formula (3) can be omitted because it has already been explained

Figure 8 should be placed just after Line 252 and Figure 9 just after Line 260. Currently, these two drawings take up too much space

Figure 8 - As described in the text of the article, the vertical axis should be pn and meanwhile there is a mysterious unexplained magnitude p, while pn is given as a parameter. Please explain this more closely.

Line 275 - The article does not explain where the fire velocity values vn were taken, which are one of the most important factors affecting the entire process.

Line 283 – This line included subtitle can be deleted.

Line 287 – value mwl has not been sufficiently explained.

Line 301 - after the first sentence in this line, it is advisable to place the entire description contained in the title of Table 2, of course, except the title itself. Table 2 should be placed just after Line 301. After appropriate reduction it can be given horizontally in the text. It will then take up less space without losing anything of its readability.

Lines 311-314 – no sources has been given for this text.

Line 342 - Table 3 should be placed just after Line 342. It is advisable to move the text from the table description (except the title itself) to this line.

Line 351 – Figures 10 and 11 should be placed just after this line.

Figure 10 – I would suggest reorganizing Figure 10 into three separate double figures 10, 11 and 12 placed horizontally each corresponding to a different fire (Avinyo, Blanes, Sant Fruitos de Bages), in which the actual burnt area would be on the left and the simulated burnt area on the right. Of course, this would require appropriate changes in the text and description of the drawings.

Line 360 - Figures 12 and 13 should be placed just after this line.

Line 364 - The designation of the indicator function of the real burnt area in the text differs by the subscript b in comparison with the value appearing in the formula. Are they the same quantities or different. This requires clarification or improvement of the notation.

Line 366 - The text below Table 3 should be placed directly below the formula (9). In the article does not explain sufficiently why the value "1" is found under the integral in the formula.

Line 374 - Table 4 should be placed just after Line 374.

Table 4 – The second sentence in title of the table can be deleted because the same can be found in the main text.

Lines 506-507 - this piece of text does not match the template.

Author Response

The Authors

Reviewer 4 Report

The manuscript entitled “PROPAGATOR : an operational cellular-automata based wildfire simulator” describes a system to simulate wildfires in real time starting from ignition point, wind, humidity, orography and vegetation type. Output is supplied as a probabilistic map where the spatial distribution of fire probability is shown pixel by pixel. Five case studies located in the Mediterranean area, three in Catalonia (Spain) and two in Italy, are presented.

Results apparently could be of great interest for an operational use, but a detail undermines the work in its fundamental basis. The underlying problem affecting all the outcomes involves the erroneous way in which simulations have been conducted, that are clearly inconsistent with the real time forecast of the final burnt area perimeter: “ignition point were assumed starting from the environmental conditions and the final burnt areas”, therefore only after the conclusion of the event the system is able to reproduce the final perimeter of the burnt area with the showed reliability. The location of ignition points could deeply influences the final perimeter of the burnt areas so this is not a minor negligible aspect. At the same way the variation over time of the wind can be known before the end of the event only if the system is not coupled to a forecasting weather model. Thus, if it is chosen to maintain the present general frame, a deep renovation is required in order to reach the declared goal of the study: to carry out and evaluate new simulations conducted with the real ignition point and with the constant wind observed at the starting time of the fire event. Is not foregone that from the new results will descends the same conclusion.

Also other problems affect the paper: it is not possible a satisfactory evaluation the work because of lack of some central information, therefore some points should be clarified in order to adequately estimate the goodness of the work.

Some other presented information in my opinion are not of any importance for a scientific work and should be completely removed from the manuscript, it is the case of paragraph “1.3. The synopsis of PROPAGATOR implementation: history of the development of an operational and easy-to-use simulator”.

Primarily there are three obscure methodological points.

In order to reproduce the results, is of crucial importance describe how the software generates each one of the 100 single stochastic realization used to obtain the ratio of fire frequency.

Secondly is not clear at all how the authors justify eq (1) starting from the cumulative binomial probability, in particular should be explained the logical and mathematical steps for the final form and the way as the modifications caused by wind, orography and fuel moisture content are introduced.

The last issue involves the performance indicator 1 and 2. How is defined the integration domain? Depending on this definition false positives and/or false negatives are considered or not. Consequently to this choice it would be appropriate to introduce a forth indicator to rend explicit the false positive. It would be better also to provide at least someone of the common statistical indexes such as K index or Sorensen index.

To achieve an actual visual comparison of simulations with observed perimeter is necessary to create an image where the two perimeters overlap.

Another issue regards the Table 4 where values appear extremely low compared with related images and so definitely not compatible with reliable values. For example the I3 indicator value for Ittiri fire means that only less then 1‰ of pixels are false negatives, but comparing the simulation with the actual perimeter (just only in the northern-east part of the image and the bulge in the west direction in the right part of the image are about 200ha of false negatives) this does not seems the case. It could be guess a two order magnitude higher error.

It is important underline as Arca et al 2019 (https://doi.org/10.1071/WF18078) analyze and reconstruct the same wildfire of Ittiri where the “extensive suppression interventions conducted on the fire flanks during the earliest phase of fire propagation” avoided that the fire reached the village. In this case a southern wind derived from a 1.5 km resolution meteorological model was used to initialize the simulation, and it looks like more compatible with fire suppression attack on flanks than the south west wind direction used in the in the manuscript, even more if it is considered that in the present work wind direction comes from a meteorological station located at least 10 km.

For reasons of clarity and to reproduce the results I think should be indicated, even in the map, the locations of the meteorological stations which supplied the wind observations used in the simulations.

A second issue is a lack of statistical evidence to assert that the model “confirms the success of an operational tool”, five cases are not enough. To justify such a statement it should be considered a general validation comprising a larger number of cases.

In the manuscript is not found the value of nominal fire spread velocity, which represents an important model parameter, thus is needed to provide such a information. To improve readability all information on “nominal fire spread probability” and “nominal fire spread velocity” sparse in the manuscript should be grouped together.

Therefore a number of important revisions across the manuscript are needed and some points should be radically changed, clarified or added in order to accept this manuscript for publication. Anyway the work could have the characteristics to be published after a major revision.

Specific technical suggestions to improve the manuscript are listed below.

Page 1, L20-26: Please rework the sentence in order to improve readability.

Page 2, L36: it does not seems a “new behavior”.

Page 2, L48: “as the spatial segmentation into polygons of fire potential introduced by Castellnou et al in [5]”, please supply a wider explanation or remove it from the text.

Page 3, L88: “they can be integrated in the framework of existing databases with relative ease”

Page 3, L91: “(ii)... “

This is not a typical characteristic of the Grid-based stochastic model, also the other two kind of model can be implemented in the same way with the same ease.

Page 3, L111: Please remove this paragraph that nothing add to the scientific content of the work.

Page 7, capture of Figure 3: please restructure the second sentence.

Page 7, L183-193: define and supply a description of “nominal fire spread probability” and of “nominal fire spread velocity”. The variable name pn and vn should also be introduced by this point, together with all other information sparse into the manuscript. Is important underline as there is no reference in the manuscript to the numerical values of vn introduced in the model, please supply this information.

Page 9, L227-228: ‘ “Not fire-prone forest” class represents the low-flammable forests: its probability of being

ignited is quite low, except if the burning cell is a “Fire-prone conifers” cell ’ description does not corresponds to values presented in Table 1

Page 10, L249: substitute “direction and the” with “when”

Page 10, L255: “Spread Probability can thus vary in a range between at about zero and 0.7”, to improve readability this passage should be made more explicit

Page 10, L257-258: em should be briefly described, saying at least as it is derived, and that em ∈[0,1]

page 11, Figure 9: values in the horizontal axes are not expressed in % such as is stated in the capture.

Page 15, eq(7): please define set S of the integration domain, and the integrand function Ib(x,Tend)

Page 15 par. 4.1: please supply a more detailed description of all the performance indicators.

Page 18 pL390-392:”the good overall performance obtained confirms the success of more than a decade of continuous scientific and technical improvement of an operational tool conceived for practitioners and decision-makers”: five analyzed case allow to consider the study only a preliminary study, demanding to a wider sample of cases a general validation of the model, so in any case it cannot still be considered an help for practitioners and decision-makers.

Author Response

Plese see the attachment.

The Authors

Round 2

Reviewer 3 Report

Although the work has been largely improved according to my suggestions, I still have a few reservations about it, including one more serious one. They concern Chapter 6, which practically contains no conclusions, only a summary of the research. So please, put in it some ordered and numbered general conclusions resulting from the simulation tests.

Although Latex is a good tool, its automatism is not always beneficial, because the logical arrangement of tables and drawings at work is quite important.

Please also correct the formulas (7), (8) and (9), in which instead of the quantity x there should be the xp (it has been previously corrected).

Author Response

The Authors

Reviewer 4 Report

A comparison of the first and second version evidences a clear amelioration of the work , now it is possible to find all information to reply the study.

On the other hand not all the problems highlighted in the first version are solved.

A confused validation strategy still persist.

If an operational use is designed, validation should involves simulations initialized with the data available in operational conditions, i.e. at the ignition moment, not at the end of the real event. At line 331 of the manuscript can be read “"Wind speed and direction" are taken as significant for the entire fire event,while in the simulation more detailed wind conditions have been implemented”. Given that wind is considered constant over space and that “actual meteorological condition had been used for all the simulations”, does this “more detailed wind conditions “ refers to variation over time?

In this case, as underlined in the previous review, remains still unknown how the wind variation over time can be available at the beginning of the simulation without considering any kind of forecast. It is so clear that the presented simulations could not be produced in a actual real time. So the author aims to valuate a real time product with an output that is not possible generate during an event: it could clearly bring to evaluation that can easily fall in a false right response of the system, resulting in a overestimation of the capacity of the system to forecast in real time fire behavior. Considering for example the observed data supplied in the cover letter, wind intensity pass from 11m/s at the ignition time to 4m/s after 10 hours. Initializing the simulation with this set of data the starting rate of spread slow down significatively, avoiding an over estimation of the burnt surface surely experienced with a simulation initialized with the constant wind taken at the ignition time. A real time simulation taking into account this decrease is possible only if in real time a future value of wind intensity is available. So this issue is not a barren and abstract methodological problem of no practical interest. I hope no more words are needed to state as the adopted procedure is not correct. Therefore, keeping the same general frame, new simulations with constant wind over time are needed.

I am agree with the Author that the Ittiri fire is not the main point of the discussion, there is in the manuscript a more important problem that should be solved. But could be worth to note how, even if could could be plausible in some circumstances affirm that “it is really difficult to sustain that the output of a WRF model is better than a meteorological observation 10 km far from the ignition point in a quite homogeneous topography”, this is not surely the case. The author can easily verify diffused slope of 40-50% in the domain (for example nearby the fire fighting intervention on the left flank of the fire or nearby the top of the hill where the is located the meteorological station used to acquire the wind), and the author can not disagree that this is not properly a homogeneous topography. Furthermore considering that the meteo station is positioned at 700m asl on the top of a hill, using those data to estimate the wind in a location more than 500m lower in elevation, located 15-20km away, in a quite complex orography could be considered at least a challenge not a reliable and safe estimate. Also appear questionable the argument by which differences in direction and intensity between the wind simulated with COSMO model and observed data are not significative. It should be also clarified why between the great number of station surrounding the fire, some of them even located in a closer position (http://www.sar.sardegna.it/documentazione/strumenti/retestazioni.asp), a “suspect station” was selected (http://www.sardegnaambiente.it/documenti/21_393_20200204130013.pdf, p. 19).

If the author is aiming for a reliable instrument, should be remarked as reliability is based on scientific evidence, therefore is out of doubt that, being validation not an act of faith, the “hundreds of wildfire occurrences, that varied significantly in vegetation pattern, burnt area, wind conditions, orography and human intervention level” cited in the manuscript should pass through statistical evidence for proving the capacity to supply information on fire behavior in real time. At the same way, the fact that the system “has been already tested operationally by several end users during the ANYWHERE [30] demonstration phase” cannot be intended as an indicator of reliability till the moment in which a peer review of the analysis is carried out even though if in some events it gave an important help to the fire management. Consequently, from the scientific point of view of this journal, at the moment it could be considered an appreciable and very interesting preliminary study with a promising capacity of reproduce the fire behavior in a variety of situation, having all the characteristic for being published if some aspects will be adjusted. Just to speak of concrete aspects, for Civil Protection and fire managers it is of primarily importance avoids fire entrapment which happens when wind suddenly change direction and cause the fire fighters to find closed all the way out to escape from a fire line out of control. (https://www.wildfirelessons.net/orphans/viewincident?DocumentKey=cee6f970-4fe8-40f5-8ae2-2f9a991873b0, https://www.tandfonline.com/doi/abs/10.1080/10807039.2013.871995?scroll=top&needAccess=true&journalCode=bher20) This kind of events, that PROPAGATOR is not able to simulate because it is not coupled with any kind of meteo forecast, generate the main part of victims, particularly in the fire fighters. So before state that a fire simulator “confirms the success of more than a decade of continuous scientific and technical improvement of an operational tool conceived for practitioners and decision-makers” the above mentioned aspect, that definitively represents the most important one, grabbing the main part of the effort of the fire fight in terms of strategic and tactical intervention, should be considered.

I remain of the same opinion about the paragraph 1.3, where the very few scientific and technical supplied information could be inserted into the other parts of the manuscript.

Author Response

The Authors
